# Rapid Classification and Quantification of Camellia (*Camellia oleifera* Abel.) Oil Blended with Rapeseed Oil Using FTIR-ATR Spectroscopy

**DOI:** 10.3390/molecules25092036

**Published:** 2020-04-27

**Authors:** Jianxun Han, Ruixue Sun, Xiuying Zeng, Jiukai Zhang, Ranran Xing, Chongde Sun, Ying Chen

**Affiliations:** 1College of Agriculture & Biotechnology, Zhejiang University, Zijingang Campus, Hangzhou 310058, China; hanjianxun19830418@163.com; 2Agro-Product Safety Research Center, Chinese Academy of Inspection and Quarantine, Beijing 100176, China; zhjk_caiq@163.com (J.Z.); xingranrancaiq@163.com (R.X.); 3College of Food Science & Nutritional Engineering, China Agricultural University, Beijing 100083, China; 18611907955@163.com; 4Scientific Research Department, Ganzhou Quality Supervision and Inspection Institute, Ganzhou 341000, China; shanshuihuscau@163.com

**Keywords:** camellia oil, adulteration, FTIR, chemometric, PCA, LDA, PLSR

## Abstract

Currently, the authentication of camellia oil (CAO) has become very important due to the possible adulteration of CAO with cheaper vegetable oils such as rapeseed oil (RSO). Therefore, we report a Fourier transform infrared (FTIR) spectroscopic method for detecting the authenticity of CAO and quantifying the blended levels of RSO. In this study, two characteristic spectral bands (1119 cm^−1^ and 1096 cm^−1^) were selected and used for monitoring the purity of CAO. In combination with principal component analysis (PCA), linear discriminant analysis (LDA), and partial least squares regression (PLSR) analysis, qualitative and quantitative methods for the detection of camellia oil adulteration were proposed. The results showed that the calculated I_1119_/I_1096_ intensity ratio facilitated an initial check for pure CAO and six other edible oils. PCA was used on the optimized spectral region of 1800–650 cm^−1^. We observed the classification of CAO and RSO as well as discrimination of CAO with RSO adulterants. LDA was utilized to classify CAO from RSO. We could differentiate and classify RSO adulterants up to 1% *v*/*v*. In the quantitative PLSR models, the plots of actual values versus predicted values exhibited high linearity. Root mean square error of calibration (RMSEC) and root mean square error of cross validation (RMSECV) values of the PLSR models were 1.4518–3.3164% *v*/*v* and 1.7196–3.8136% *v*/*v*, respectively. This method was successfully applied in the classification and quantification of CAO adulteration with RSO.

## 1. Introduction

*Camellia oleifera* Abel. (*C. oleifera*) is an important woody oil crop that is widely cultivated in Southeast Asia and Southern China including Jiangxi, Hunan, Hubei, Zhejiang, Guangxi, and Guangdong provinces. Camellia oil (CAO) is extracted from the seeds of *C. oleifera* by cold pressing, solvent extraction, or aqueous enzymatic treatment [1], and has drawn much attention owing to its nutritional benefits. Versus common edible oils such as soybean oil (SBO), rapeseed oil (RSO), corn oil (CO), and sunflower oil (SFO), CAO contains high oleic acid concentration (almost 70–86% of its fatty acid composition) [1,2]. CAO is also rich in vitamin E [3] and has high total phenolic, α-tocopherol, and squalene contents [4,5]. Furthermore, CAO has unique therapeutic properties such preventing coronary heart disease, inhibiting atherosclerosis, and reducing blood pressure and cholesterol [6]. CAO was first recorded as a medical oil in the Compendium of Materia Medica and is recommended as a healthy edible oil by the Food and Agriculture Organization (FAO) [7].

Due to its valuable composition and related properties, CAO is gaining popularity as a food, and in pharmaceuticals and cosmetics. These oils are relatively expensive because of their nutritional values, and thus it is important to protect them from adulteration with low price/quality edible oils such as RSO, SBO, SFO, CO, and peanut oil (PO) [8,9,10,11]. Moreover, rapeseed and derivatives such as RSO are recognized as a new potential source of allergies [12]. Therefore, the development of rapid and accurate analytical methods for the determination of CAO authentication is helpful to protect consumers.

To probe the authenticity of CAO, a range of analytical methods based on gas chromatography (GC) [13], gas chromatography-mass spectrometry [14], liquid chromatography-mass spectrometry [15], nuclear magnetic resonance spectroscopy [9], high-performance thin layer chromatography [16], and stable carbon isotopes [17] have been investigated to differentiate and detect CAO among other vegetable oils. These have proven to be sensitive, selective, and accurate for CAO authentication. However, some of these techniques are relatively expensive, labor intensive, and require sample modification as well as the use of pollutants as solvents. Therefore, improvements in noninvasive, less resource-intensive, inexpensive, and rapid methods are important for safeguarding consumers, industry, and retailers.

Recently, Fourier transform infrared (FTIR) spectroscopy has received much research attention in the authentication of edible oils. FTIR is simple, rapid, and requires minimal sample preparation. To date, several FTIR methods have been reported to differentiate edible oils and determine adulteration levels. They have been used to discriminate olive oil adulterated with various vegetable oils [18,19,20]; cold-pressed sesame oil (SO) adulterated with hazelnut oil, RSO, and SFO [21]; cold-pressed black cumin seed oil adulterated with SFO, hazelnut, cotton, grape seed, and virgin olive oils [22]; mustard oil adulterated with argemone oil [23]; virgin coconut oil adulterated with paraffin oil [24]; neem and flaxseed oil adulterated with vegetable oils [25]; walnut oil adulterated with SFO [26]; hempseed oils from different geographic origins [27]; and extra virgin olive oils from whole and stoned olive pastes [28]. All of these reports used FTIR spectroscopy to monitor the adulteration of edible oils. Although there have been several studies on near-infrared (NIR), mid-infrared (MIR), Raman, and fluorescence spectroscopy investigations into camellia oil adulteration [10,11,29], studies on the authentication of camellia oil using FTIR spectroscopy remain limited. 

As a huge number of informative and uninformative variables can be contained in FTIR data, it is necessary to conduct wavelength selection or variable extraction before the classification procedure. Principal component analysis (PCA) is the classical and the most widely used dimensionality reduction method [30]. It is an unsupervised learning algorithm used to explore the variability in a dataset and evaluate whether different groups of samples exist when the dimensionality of the data decreases [31]. It helps to identify the most important variables and eliminate uninformative spectral variables in the data set that can contribute to a more robust and less complex model [32]. Linear discriminant analysis (LDA) is a statistical method used to find a linear combination of structures with the potential to characterize or separate classes of objects or observations. For qualitative analysis, principal components contributing to the variance of the dataset were subjected to LDA in an attempt to predict the likelihood of a sample belonging to a previously defined group [33]. Multivariate calibration is a useful chemometric method for analysis of complex mixtures as it enables the rapid and simultaneous determination of each component in the mixture [34]. Partial least squares regression (PLSR) is a well-known factorial multivariate calibration method that decomposes spectral data into loadings and scores, building the corresponding calibration models from these new variables [34,35]. It is widely used in quantitative prediction methodologies based on spectroscopic data. In PLSR models, the concentration of analytes is correlated with the principle components or factors, which are linear combinations of the original spectral variables.

Here, we developed FTIR spectroscopic methods associated with chemometrics analysis for rapid discrimination between CAO and six other vegetable oils (RSO, SBO, CO, SFO, PO, and SO) as well as for quantification of the degree of CAO adulteration with RSO.

## 2. Results and Discussion

### 2.1. FTIR Spectra of Edible Oils

The FTIR fingerprint spectra of seven brands of CAO and another six vegetable oils (seven brands of SBO, seven brands of CO, seven brands of RSO, seven brands of PO, seven brands of SFO, and seven brands of SO) in the mid-infrared region of 4000–650 cm^−1^ are shown in Figure 1. For edible oils, most peaks and shoulders in the spectrum were attributable to specific functional groups. The details of functional groups responsible for FTIR absorption bands are described in Table 1. In our study, the spectra of these oil samples showed many similarities among absorbance bands, which is consistent with previous reports [10,27,30,36,37,38]. All plant oils are made up of triacylglycerol (92%), low concentrations of di- and mono-acylglycerols (5%), and low levels of other components [23]. The precise positions and shapes of the absorption bands are affected by oil composition [33]. GC is a routine method in fatty acids profiling. From GC analysis, fatty acid profiles of these oils were in the range described by Chinese national standards (GB/T 11,765, GB/T 1535, GB/T 19,111, GB 1536, GB/T 10,464, GB/T 1534, and GB/T 8233), as shown in Table 2. It was found that these oil samples were pure. In addition, the results showed that there were some differences in the fatty acid profiles of CAO and other oil samples, especially in the content of monounsaturated and polyunsaturated fatty acids. Therefore, variations in fatty acid profiles led to a discrepancy in the FTIR spectra, which may contribute to the classification of edible oil types and the detection of adulteration levels.

One report mentioned spectral differences between pure CAO and cheap oils despite different absorbance units in the peaks (RSO, SFO, CO, and PO) that were difficult to distinguish [11]. Here, we closely compared the signals of all absorbance bands. We found that there were obvious opposite characteristics for two adjacent bands of 1119 cm^−1^ and 1096 cm^−1^ attributed to C–O stretching (ester group) among these oils. In pure CAO, the band height of 1119 cm^−1^ was higher than that of 1096 cm^−1^, which is the opposite of other pure oils (RSO, BSO, CO, RSO, PO, and SO). This has also been found between extra virgin olive oil (EVOO) and canola oil [38], as well as olive oil with SFO, CO, and RSO, and cottonseed oil [42]. These differences can be exploited for classification of CAO and the other six vegetable oils. Ratios between absorbance values, instead of absolute absorbance values, are used, because, in the former, all variable circumstances involved in the sample preparation of each spectra are eliminated [43]. The intensity ratio of these bands (I_1119_/I_1096_) was also calculated, as shown in Figure 2. Some fluctuations in the I_1119_/I_1096_ ratio were observed among samples with CAO, RSO, PO, SO, CO, SFO, and SBO, giving I_1119_/I_1096_ ratios of 1.045–1.083, 0.961–0.981, 0.973–0.988, 0.935–0.961, 0.931–0.944, 0.905–0.922, and 0.905–0.918, respectively. These fluctuations in the I_1119_/I_1096_ ratio among different oils might be due to variations in the plant sources or preparation processes. The spectra changes between study samples are presented with the error bars on the spectra (Appendix A). These results clearly showed that I_1119_/I_1096_ ratios greater than 1.000 would only be obtained from pure CAOs, while values lower than 1.000 would be obtained from other edible oils. Therefore, the intensity ratio of I_1119_/I_1096_ is a potential tool for initially distinguishing pure CAO from non-CAO.

RSO is easily available and relatively cheap. Its oleic acid content is much closer to CAO, as shown in Table 2, and it is more likely to be used as an adulterant in CAO. To investigate the change in CAO/RSO spectra, one brand of each of these oils was used to prepare binary blend oil samples at different concentrations including 1%, 3%, 5%, 10%, 15%, 20%, 25%, 30%, 35%, 40%, 45%, 50%, 60%, 70%, 80%, and 90% *v*/*v*. The spectral change in different concentrations of CAO/RSO admixtures is shown in Figure 3. It can be seen that the FTIR intensity at 1119 cm^−1^ and 1096 cm^−1^ changed concomitantly with the ratio of adulterated oil in the binary mixture. As the percentage of adulterated oil increased, the FTIR intensity at 1119 cm^−1^ became weaker, while the intensity at 1096 cm^−1^ became stronger. Furthermore, the I_1119_/I_1096_ ratio decreased with increasing adulterated oil content (Table 3). We found that the intensity ratio of I_1119_/I_1096_ was correlated with the concentration of BSO adulterated in CAO (Appendix A). When the concentration of CAO/RSO admixtures was between 0% and 50%, the peak height of 1119 cm^−1^ was higher than that of 1096 cm^−1^ with a corresponding value of I_1119_/I_1096_ greater than 1.000, whereas the results were reversed when the concentration of CAO/RSO admixtures was between 60% and 100% (Appendix A). Consequently, we could rapidly and accurately determine the adulteration by the naked eye if the adulteration level of RSO in CAO was above 50% *v*/*v*. In contrast, it was hard to identify adulteration at lower concentrations. Accordingly, the application of chemometric analysis using full or selected spectral regions was necessary to determine the sample authenticity and the adulteration level.

### 2.2. Principal Component Analysis

PCA is used to determine natural clustering of samples and discriminant function analyses to assess the source discrimination potential of features based on the differences of their origin (namely commercial brands) and fatty acid profiles. PCA can also identify the most important variables and eliminate uninformative spectral variables in the data set. Thus, PCA highlights a more robust and less complex model to predict sample adulteration. Samples are considered similar if they lie closer in the score plot, while dissimilar samples lie apart from each other [44]. The spectra of CAO, RSO, SBO, CO, SO, PO, and SFO show that there were some low signal-to-noise regions in the full spectrum (4000–650 cm^−1^) but more complicated absorbance profiles in the fingerprint region (1800–650 cm^−1^). To achieve optimal discrimination, PCA was applied to four different spectral regions (4000–650 cm^−1^, 3050–2750 cm^−1^ and 1800–650 cm^−1^, 3050–2750 cm^−1^, 1800–650 cm^−1^) to investigate the similarities and differences between CAO and other vegetable oils.

Figure 4a represents the three-dimensional (3D) score plot obtained from PCA using the full spectral range of 4000–650 cm^−1^. The first principal component (PC1) explained 69% of the variance, while the second (PC2) and the third (PC3) explained 4% and 3%, respectively; therefore, approximately 76% of variance could be described by the three principal components. Combining spectral ranges of 3050–2750 cm^−1^ and 1800–650 cm^−1^, PCA was completed with three PCs accounting for 83% of the explained variance (PC1, 74%; PC2, 5%; PC3, 4%) (Figure 4b). In Figure 4c, PCA was constructed in the region of 3050–2750 cm^−1^. Three PCs of the model accounted for 74% of the explained variance (PC1, 66%; PC2, 5%; PC3, 3%). Another three PCs (PC1, 86%; PC2, 4%; PC3, 3%) were obtained by conducting score plots on the fingerprint region (1800–650 cm^−1^) (Figure 4d); they explained 93% of the total variance. As in the score plot of Figure 4, CAO and other vegetable oils occupied different positions; thus, PCA allows qualitative discrimination between CAO and its adulterant under analysis. However, in Figure 4a, PO, SO, and RSO were hard to separate clearly as were CO, SBO, and SFO; CO, SBO, and SFO clustered together (Figure 4b), and CO, SBO, and SFO were very close to each other, as shown in Figure 4c; therefore, they would be very difficult to distinguish using infrared spectroscopy. In Figure 4d, each oil occupied a different position, thus making these oils distinguishable from each other. Consequently, the FTIR region of 1800–650 cm^−1^ had a much better ability to differentiate the edible oils under analysis.

Using the informative spectral region (1800–650 cm^−1^), PCA was applied to determine the resemblance and disparity between CAO and its RSO adulterant in different ratios. For the sake of clarity, only pure CAO and RSO adulterated concentrations of 1%, 3%, 5%, 10%, 15%, and 20% *v*/*v* were represented in the 3D score plot; this indicated seven distinct clusters (Figure 5). In the plot, PC1 explained a variance of 91% while PC2 and PC3 explained 4% and 3%, respectively; therefore, the total variance described by the first three PCs was found to be 98%. No overlap was found between CAO and its different adulterants with RSO, which indicates that PCA allows qualitative discrimination between RSO adulterants (1–20% *v*/*v*) and pure the CAO under analysis.

The loading spectra were utilized to identify the wavenumbers that had the most influence on a particular component that was responsible for differentiation of pure CAO and its adulterants. Both positive and negative score values of a specific wavenumber loading represent the usefulness of the contribution of each wavenumber. The loading plot in Figure 6 shows an analysis of the first three PCs applied for the classification of the CAO and RSO/CAO admixtures above. The data showed that wavenumbers of 1744 (C=O stretch), 1464 (methylene C–H bending; methyl C–H deformation), 1377 (methyl C–H bending), 1155 (C–O stretch), 1160 (methylene C–H bending), 1119 (C–O stretch), 1096 (C–O stretch) and 721 cm^−1^ (cis-CH=CH bending out of plane) had relatively large absolute score values, which indicated that these wavenumbers were the most important wavenumbers for the formation of principal components. Therefore, these major wavenumbers could be utilized for further classification and differentiation of CAO from its RSO adulterants.

### 2.3. Linear Discriminant Analysis

LDA is usually used to characterize or separate classes of objects or observations and to predict the likelihood of a sample belonging to a previously defined group. LDA cannot be utilized on entire sample groups because there are usually more variables in the data matrix than sample numbers; this is a common problem in the case of chemometric (LDA) analysis of different food matrices [45]. For classification and discriminative analysis of virgin coconut oil (VCO) adulteration with paraffin oil (PO), Amit et al. established an LDA model developed with a subset of samples (VCO 100%, PO 1% and PO 3%) [24]. The LDA was constructed based on 13 selected wavenumbers (2954, 2924, 2852, 1743, 1465, 1417, 1377, 1228, 1155, 1111, 962, 872, and 721 cm^−1^) from the loading spectra of PCA. This group found that LDA could differentiate and classify PO adulterants up to 1% *v*/*v*. By combining absorbances at the 12 wavenumbers (3008, 2922, 2853, 1743, 1464, 1417, 1376, 1238, 1160, 1118, 1098, and 722 cm^−1^), Jamwal et al. used LDA for classifying mustard oil (MO) from argemone oil (AO) on a simplified data set (MO, AO 1% and AO 5%) [23]. They found that the least detectable percentage of AO in MO was the least adulteration level studied (1% *v*/*v*).

To assess the classification ability of CAO adulteration with RSO, we developed an LDA model with a subset of samples constituted by CAO adulterated with 10%, 5%, 3%, and 1% *v*/*v* of RSO and pure CAO. The LDA was constructed based on the seven selected wavenumbers (1744, 1464, 1377, 1155, 1119, 1096, and 721 cm^−1^) mentioned in the loading plot (Figure 6). The sample subset of the calibration set included 15 replicates of every group. Figure 7 shows the similarity plot as defined by the first two discriminant factors that had the most important role in the classification of CAO and its adulterants. The groups with RSO 10%, RSO 5%, RSO 3%, RSO 1% *v*/*v*, and pure CAO were clearly separated and positioned at different regions; there was no overlap. The confusion matrix obtained from LDA showed that all of the initial groups were 100% classified (Appendix A). 

In cross-validation, each group is classified by functions derived from all other cases other than the case in the analysis. In Table 3, 100% of the grouped cases in cross-validation were correctly classified. No misclassification of the samples was found when cross-validated (Appendix A). The results show that the group of RSO 1% belong to the lowest concentration of RSO used. Thus, 1% *v*/*v* was the lowest detected RSO concentration in CAO. One report described a soft independent modeling of class analogy (SIMCA) technique used to discriminate camellia oil from camellia oil adulterated with soybean oil based on MIR spectroscopy. This technique had a detection limit of 5% (*w*/*w*) [10], which was higher than the values obtained in the present study. Therefore, the LDA model developed here is suitable for the discrimination of CAO adulterated with RSO.

### 2.4. Quantitative Analysis of CAO Adulteration with RSO Based PLSR

PLSR is a well-known regression methodology for multivariate data that is mainly applied for quantitative prediction; it is usually utilized to quantify chemical parameters in vegetable oils. Moreover, this model provides enhanced results versus other regression methods for quantitative analysis of chemical parameters [46,47]. In PLSR models, the concentration of analytes is correlated with the PCs or factors, which are linear combinations of the original spectral variables. The statistical values included slope, offset, correlation coefficient, RMSEC, and RMSECV. These values are commonly applied for assessing the performance of the developed PLSR models and for stating their predictive abilities [48].

For quantitative analysis of RSO adulterated in CAO, PLSR models were constructed from the FTIR data of different binary mixing oil samples. The cross-validation method was used in a PLSR model validation with as many subsets as included in the calibration matrix (leave-one-out method) [26]. One disadvantage of the reported models to authenticate edible oils using spectroscopic techniques is the low number of different botanical species used to make the blends binary of adulterated oils with other vegetable oils [49]. Therefore, we selected five brands of CAO (*n* = 5; CAO–1 to 5) and three brands of RSO (*n* = 3; RSO–1 to 3) to fabricate oil blends as more relevant calibration models. In the scope of this work, the RSO blended concentration spanned a range of 1–50% (*v*/*v*) in CAO because adulterated CAO can be easily detected by the intensity ratio of I_1119_/I_1096_ when the concentration of RSO exceeds 50%. In our study, pure CAO (5 samples) and CAO/RSO-adulterated samples (180 samples) were selected, and the resulting models were used to calculate the concentration of unknown samples (Appendix A). 

Table 4 showed a summary of the figures-of-merit of 15 multivariate calibration models in terms of slope, offset, R^2^ value, RMSEC, and RMSECV. A perfect PLSR model would have a slope of 1, an R^2^ value of 1, and an offset of 0 [25]. In this work, the slopes of all samples were 0.9703–0.9969 and were close to 1. The R^2^ and RMSEC values of the PLSR calibration models were 0.9836–0.9969 and 1.4518–3.3164% *v*/*v*, respectively, while the R^2^ and RMSECV values in the validation models were 0.9805–0.9959 and 1.7196–3.8136% *v*/*v*, respectively. The offset values for the adulterant level prediction were 0.0606–1.3360. The high R^2^ value indicated that good linear relationships were calculated for all the calibration models. The RMSEC and RMSECV values represented the spread of calibration and validation data points along the fitted line with the best results obtained when these two values are similar [50]. The values of RMSEC and RMSECV obtained in this work indicated that the actual and predicted values were similar. On the basis of the developed PLSR models, the lowest limit of quantitative detection of RSO in CO could be determined to be 1% *v*/*v*. Therefore, predicting adulterant oil concentration in CAO might be possible using the proposed PLSR models. These results suggest that FTIR coupled with chemometric regression methods, such as PLSR, are very effective tools for detecting CAO adulteration with RSO at low adulteration levels.

## 3. Materials and Methods

### 3.1. Chemicals and Samples

Mix standard of fatty acid methyl esters (FAMEs) of 37 compounds was purchased from Sigma-Aldrich Chemicals (Bellefonte, PA, USA). Spectroscopic grade ethanol was purchased from Merck (Darmstadt, Germany). Distilled water was produced by a Millipore Water Pro water system (Millipore, Molsheim, France). A total of 49 vegetable oils including camellia (*n* = 7; CAO–1 to 7, from seven different brands), rapeseed (*n* = 7; RSO–1 to 7, from seven different brands), soyabean (*n* = 7; SBO–1 to 7, from seven different brands), corn (*n* = 7; CO–1 to 7, from seven different brands), peanut (*n* = 7; PO–1 to 7, from seven different brands), sunflower (*n* = 7; SFO–1 to 7, from seven different brands), and sesame oils (*n* = 7; SO–1 to 7, from seven different brands) were collected. To diversify the sampling population, the CAOs were collected from seven provinces in China including Jiangxi, Hubei, Zhejiang, Guangdong, Guangxi, Fujian, and Anhui, which represent the main areas in which *C. oleifera* trees are distributed in China. They were provided by Ganzhou Bureau of Quality and Technical Supervision, Jiangxi province, China. Other edible vegetable oils were provided by Jiangnan University, Wuxi, China. Detailed information can be found in Appendix A. These samples were used to construct the classification model of botanical origin. They were stored in the dark at 4 °C prior to analysis.

To investigate the spectral change of CAO adulterated with RSO, one brand of each of these oils was used to prepare binary blend oil samples. The CAO/RSO admixture samples were prepared by mixing CAO with RSO at different concentrations: 1%, 3%, 5%, 10%, 15%, 20%, 25%, 30%, 35%, 40%, 45%, 50%, 60%, 70%, 80%, and 90% *v*/*v*.

For constructing calibration models, five selected brands of CAO (*n* = 5; CAO–1 to 5) and three selected brands of RSO (*n* = 3; RSO–1 to 3) with different volume proportions of 1%, 3%, 5%, 10%, 15%, 20%, 25%, 30%, 35%, 40%, 45%, and 50% *v/v* were used to fabricate the oil blends (Table 5). Therefore, there were a total of 180 binary blend oil samples. A total of 194 samples consisting of the 180 CAO/RSO admixtures and the above 7 pure CAOs and 7 pure RSOs were used to build the quantitative PLSR model for CAO and CAO adulterated with RSO. The CAO/RSO samples were shaken for 30 min and kept at room temperature for approximately 24 h to allow homogenization of the oil mixtures.

### 3.2. Analysis of Fatty Acid Composition

The fatty acid composition of 49 vegetable oils (7 CAO brands, 7 RSO brands, 7 SBO brands, 7 SFO brands, 7 CO brands, 7 PO brands, and 7 SO brands) were measured as FAMEs using an Agilent 6820 gas chromatograph (Agilent, Shanghai, China) equipped with a flame ionization detector (FID). FAMEs were prepared according to a procedure reported previously [2].

### 3.3. Collection of FTIR Spectra

A FTIR spectrometer (PerkinElmer, Beaconsfield, England) equipped with an attenuated total reflectance (ATR) accessory with a diamond crystal module were used in this study. Infrared spectra were recorded with a resolution of 4 cm^−1^ over a wavelength range of 4000–650 cm^−1^ accumulating 16 scans per spectrum. The collection time for each oil sample spectrum was approximately 40 s. An oil sample (approximately 20 μL) was dropped onto the ATR crystal ensuring that no air bubbles were trapped on the crystal surface. Each sample was measured under identical conditions in triplicate. The crystal surface was carefully cleaned with ethanol and dried with a lens tissue after each sampling. The cleanliness was verified by collecting the background spectrum and comparing it with previous background spectra. After cleaning and drying, a spectrum of the crystal surface against ambient air was recorded and used as background for each measurement. All sample spectra were subtracted against the background air spectrum.

### 3.4. Chemometric Analysis

Multivariate data analysis was performed using PCA, LDA, and PLSR analysis using Unscrambler X10.5.1 software (CAMO Software, Oslo, Norway) and OriginPro 2018 software (Originlab, Northampton, MA, USA).

Prior to chemometric analysis of the FTIR spectral data, both Savitzky–Golay smoothing and first-order derivative were performed. The Savitzky–Golay smoothing eliminated the baseline drift and variability associated with the intensity [51]. In addition, baseline offsets may be produced in the FTIR spectra because of differences in the viscosity and composition among pure CAOs, RSOs, and CAO/RSO admixtures. These factors may influence the construction of robust models. The first-order derivative is an effective way to remove them.

### 3.5. Statistical Analysis

All measurements were carried out in triplicate, and values are expressed as the mean ± standard deviation (SD). Mean comparisons were performed using one-way analysis of variance (ANOVA). Significant differences between means were determined using Duncan’s multiple range tests. All statistical tests were done with 95% of significance and performed using SPSS 20.0 software (SPSS Inc., Chicago, IL, USA).

## 4. Conclusions

This method offers a simple, rapid, and nondestructive qualification and quantification method for the detection of CAO adulteration with RSO via ATR-FTIR spectroscopy. The calculated I_1119_/I_1096_ ratio was used to determine adulteration and monitor the purity of CAO. Combined with chemometric analysis, PCA could classify pure CAO and another six pure vegetable oils (RSO, SBO, SFO, CO, PO, and SO) and could discriminate pure CAO from those adulterated with RSO. LDA models provided excellent classification of pure CAO with its RSO adulterant. LDA could detect the lowest detectable concentration of RSO in CAO up to 1% *v*/*v*. The proportion of CAO in blends with RSO was successively quantified using PLSR models. High linear relationships were obtained from those models showing a good prediction ability with RMSEC and RMSECV values of 1.4518–3.3164% *v*/*v* and 1.7196–3.8136% *v*/*v*, respectively. Therefore, the developed methodologies based on ATR-FTIR spectroscopy were applicable to the qualitative and quantitative detection of CAO adulteration. Although blends of CAO with other vegetable oils are usually binary in real-world samples, a quality inspection department cannot identify which was used and/or how many vegetable oils were used in adulteration. Thus, higher order blends of CAO with other edible oils will be studied in future work. 

## Figures and Tables

**Figure 1 molecules-25-02036-f001:**
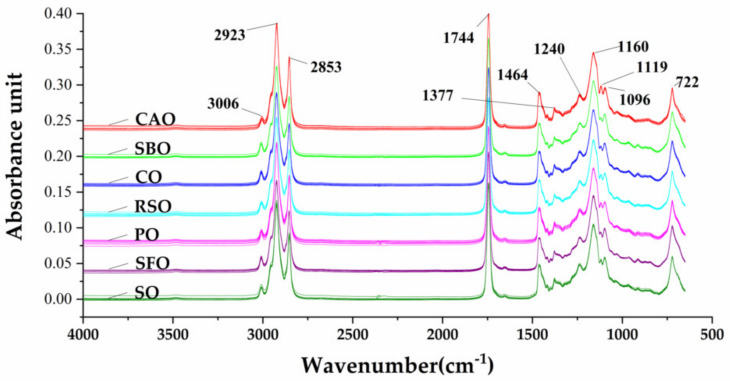
FTIR spectra of CAO, SBO, CO, RSO, PO, SFO, and SO in the full region of 4000–650 cm^−1^. CAO, camellia oil; SBO, soybean oil; RSO, rapeseed oil; CO, corn oil; SFO, sunflower oil; PO, peanut oil; SO, sesame oil.

**Figure 2 molecules-25-02036-f002:**
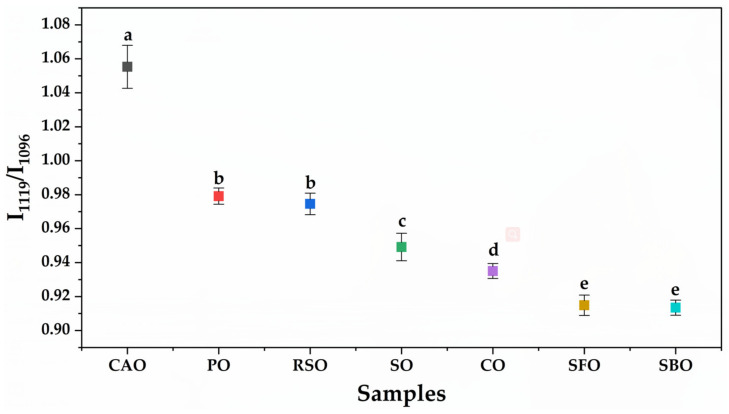
Clustered FTIR band identity ratios of I_1119_/_1096_ of different oils are shown in the plot. Different lowercase letters in the plot indicate significant differences at *p* < 0.05. CAO, camellia oil; SBO, soybean oil; RSO, rapeseed oil; CO, corn oil; SFO, sunflower oil; PO, peanut oil; SO, sesame oil.

**Figure 3 molecules-25-02036-f003:**
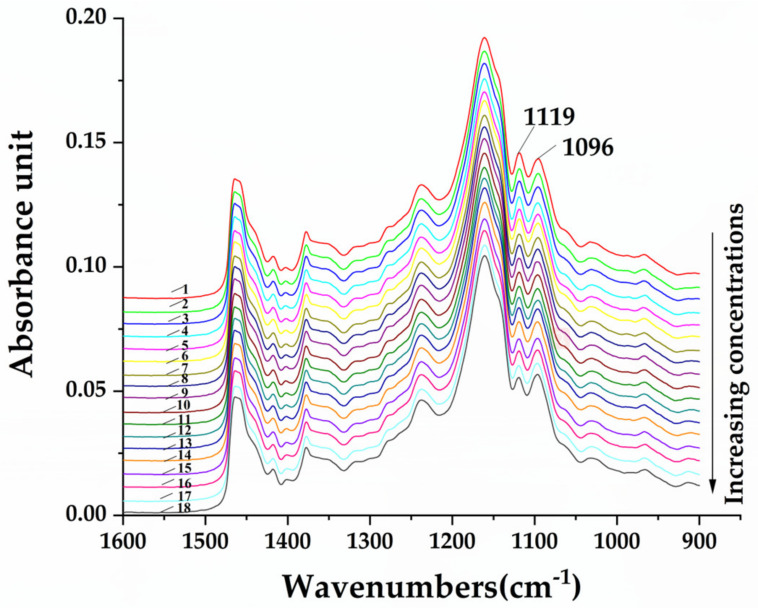
FTIR spectra of pure CAO and CAO/RSO admixtures (1–90% *v*/*v*) in the region of 1600–900 cm^−1^. 1. Pure CAO; 2. 1%; 3. 3%; 4. 5%; 5. 10%; 6. 15%; 7. 20%; 8. 25%; 9. 30%; 10. 35%; 11. 40%; 12. 45%; 13. 50%; 14. 60%; 15. 70%; 16. 80%; 17. 90%; (18) pure RSO.

**Figure 4 molecules-25-02036-f004:**
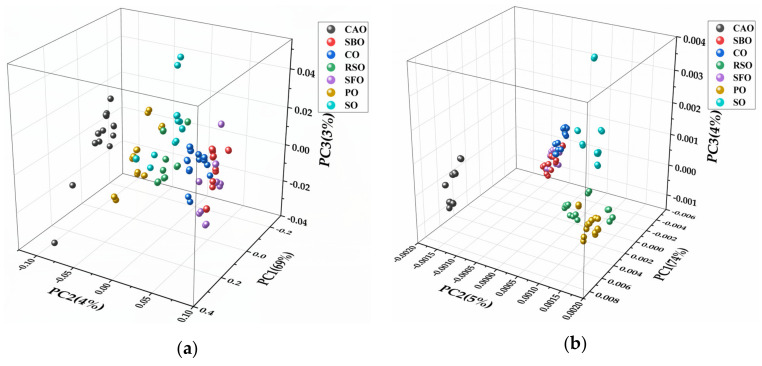
Three-dimensional score plot of pure CAO, SBO, RSO, CO, SFO, PO, and SO from principal component analysis (PCA) analysis of FTIR spectra in the wavenumber region of (**a**) 4000–650 cm^−1^, (**b**) 3050–2750 cm^−1^ and 1800–650 cm^−1^, (**c**) 3050–2750 cm^−1^, (**d**) 1800–650 cm^−1^. CAO, camellia oil; SBO, soybean oil; RSO, rapeseed oil; CO, corn oil; SFO, sunflower oil; PO, peanut oil; SO, sesame oil.

**Figure 5 molecules-25-02036-f005:**
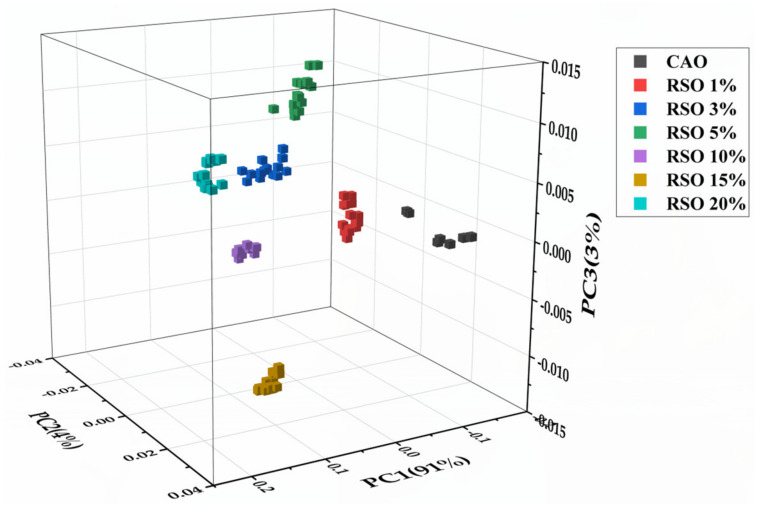
Three-dimensional score plot of CAO and CAO/RSO admixtures from PCA analysis of FTIR spectra in the wavenumber region 1800–650 cm^−1^. CAO, camellia oil; RSO, rapeseed oil.

**Figure 6 molecules-25-02036-f006:**
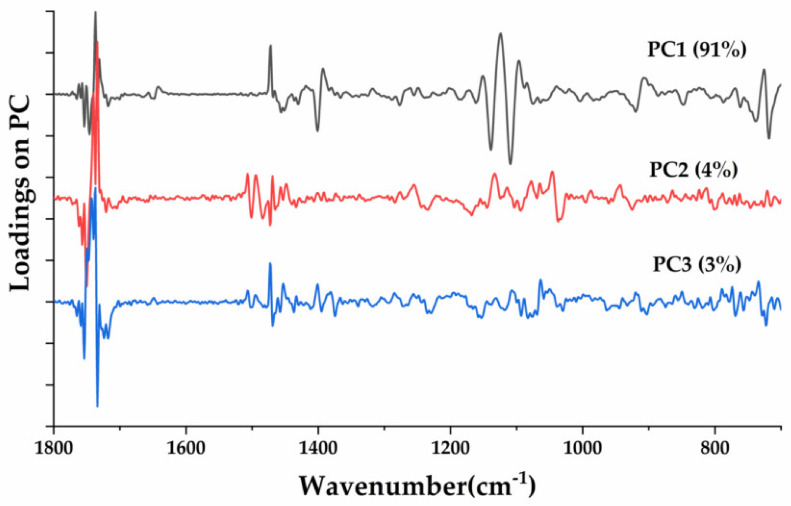
PCA loadings of CAO and CAO/RSO admixtures in the wavenumber region 1800–650 cm^−1^. CAO, camellia oil; RSO, rapeseed oil.

**Figure 7 molecules-25-02036-f007:**
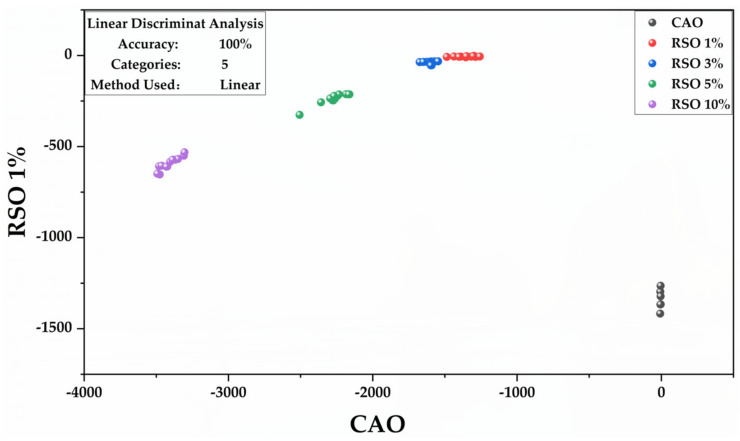
Similarity map as determined by linear discriminant analysis using the first two discriminant factors for FTIR spectral data of pure CAO and CAO/RSO admixtures (1–10% *v*/*v*).

**Table 1 molecules-25-02036-t001:** Functional groups in the edible oils and impact on the Fourier transform infrared (FTIR) spectrum: wavenumber, functional group, and mode of vibration [36,39,40,41].

Wavenumbers (cm^−1^)	Functional Group	Mode of Vibration	Possible Structural Units	Absorption Intensity
3006	=C–H (cis)	Stretching	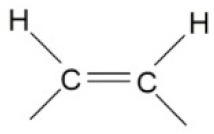	Medium
2923	–C–H (CH_2_)	Stretching (asymmetrical)	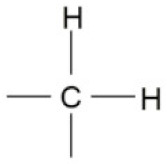	Very strong
2853	–C–H (CH_2_)	Stretching (symmetrical)	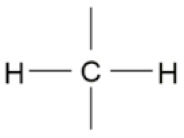	Very strong
1744	–C=O (ester group)	Stretching	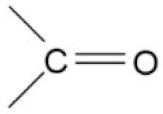	Very strong
1464	–C–H (CH_2_) –C–H (CH_3_)	Bending (scissor) and/or Deformation	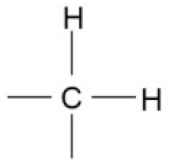 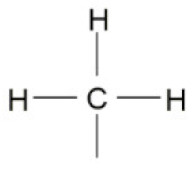	Medium
1377	–C–H (CH_3_)	Bending (symmetrical)	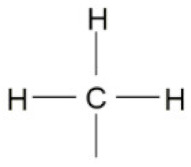	Medium
1240	–C–H (CH_2_)	Bending	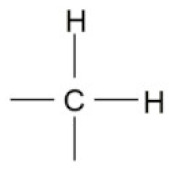	Medium
1160	–C–H (CH_2_)	Bending	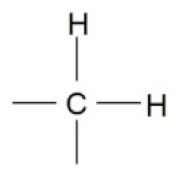	strong
1119	–C–O (ester group)	Stretching	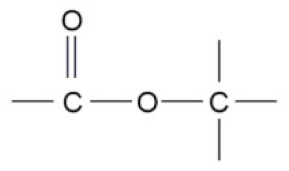	Medium
1096	–C–O (ester group)	Stretching	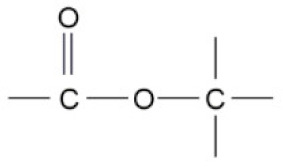	Medium
722	–(CH_2_)_n_– –HC=CH	Bending (rocking) and out-of-plane vibration	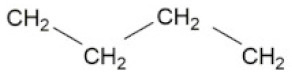 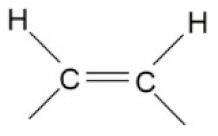	Medium Weak

**Table 2 molecules-25-02036-t002:** Major fatty acid composition of oil samples analyzed using gas chromatography.

Fatty Acids	Common Names	CAO	SBO	CO	RSO	SFO	PO	SO
C16:0	Palmitic Acid	10.44 ± 2.03	10.56 ± 0.93	13.24 ± 0.37	4.72 ± 0.22	6.43 ± 0.27	12.09 ± 0.47	9.56 ± 0.13
C16:1	Palmitoleic Acid	ND ^a^	ND	0.05± 0.01	1.17 ± 0.10	0.06± 0.01	ND	ND
C18:0	Stearic Acid	2.45 ± 0.46	3.82 ± 0.42	1.89 ± 0.05	1.98 ± 0.08	4.62 ± 0.22	3.94 ± 0.31	5.74 ± 0.27
C18:1	Oleic Acid	76.46 ± 2.72	23.17 ± 0.68	31.61 ± 0.62	64.05 ± 1.11	22.94 ± 0.29	40.87 ± 0.45	38.79 ± 0.83
C18:2	Linoleic Acid	10.25 ± 3.19	54.09 ± 0.69	51.71 ± 0.51	21.16 ± 1.23	64.68 ± 0.38	36.14 ± 0.75	45.01 ± 1.21
C18:3	Linolenic Acid	0.15 ± 0.04	7.37 ± 0.38	1.09 ± 0.14	6.29 ± 0.29	0.17 ± 0.03	0.18 ±0.05	0.44 ± 0.09
C20:0	Arachidic Acid	0.05± 0.02	0.47 ± 0.06	0.45 ± 0.02	0.08 ± 0.02	0.30 ± 0.05	1.49 ± 0.08	0.63 ± 0.05
C20:1	Arachidonic Acid	0.24 ± 0.09	0.04 ± 0.09	0.24 ± 0.09	0.24 ± 0.09	0.22 ± 0.03	1.13 ± 0.04	0.24 ± 0.09
C22:0	Behenic Acid	0.06± 0.03	0.49 ± 0.05	0.05± 0.02	0.39 ± 0.03	0.81 ± 0.05	2.85 ± 0.04	0.12 ± 0.02
C24:0	Tetracosanoic Acid	0.07± 0.02	0.04± 0.02	0.06± 0.01	0.13 ± 0.02	ND	1.47 ± 0.09	ND

Values are mean ± standard deviation. ^a^ ND represents not detected. CAO, camellia oil; SBO, soybean oil; RSO, rapeseed oil; CO, corn oil; SFO, sunflower oil; PO, peanut oil; SO, sesame oil.

**Table 3 molecules-25-02036-t003:** Confusion matrix for the classification of CAO and CAO/RSO admixtures.

		Oil Samples	Predicted Group Membership					Total
			RSO1%	RSO3%	RSO5%	RSO10%	CAO	15
Original	Count	RSO1%	15	0	0	0	0	15
		RSO3%	0	15	0	0	0	15
		RSO5%	0	0	15	0	0	15
		RSO10%	0	0	0	15	0	15
		CAO	0	0	0	0	15	15
	%	RSO1%	100.00	0	0	0	0	100.00
		RSO3%	0	100.00	0	0	0	100.00
		RSO5%	0	0	100.00	0	0	100.00
		RSO10%	0	0	0	100.00	0	100.00
		CAO	0	0	0	0	100.00	100.00
Cross-validated	Count	RSO1%	15	0	0	0	0	15
		RSO3%	0	15	0	0	0	15
		RSO5%	0	0	15	0	0	15
		RSO10%	0	0	0	15	0	15
		CAO	0	0	0	0	15	15
	%	RSO1%	100.00	0	0	0	0	100.00
		RSO3%	0	100.00	0	0	0	100.00
		RSO5%	0	0	100.00	0	0	100.00
		RSO10%	0	0	0	100.00	0	100.00
		CAO	0	0	0	0	100.00	100.00

**Table 4 molecules-25-02036-t004:** Calibration and validation results of the partial least squares regression (PLSR) models for the mixtures of RSOs in CAOs from 1800–650 cm^−1^.

Oil Admixtures		Number of Factors	Calibration Set				Validation Set			
			Slope	Offset	RMSEC (% *v*/*v*)	R^2^	Slope	Offset	RMSECV (% *v*/*v*)	R^2^
RSO-1 (0–50%)	CAO-1	4	0.9836	0.4447	3.3164	0.9836	0.9703	0.8635	3.8136	0.9805
	CAO-2	4	0.9969	0.0852	1.4518	0.9969	0.9901	0.2067	1.7196	0.9959
	CAO-3	4	0.9964	0.0978	1.5558	0.9964	0.9920	0.2268	1.8913	0.9951
	CAO-4	4	0.9952	0.1279	1.7785	0.9953	0.9880	0.3294	2.1492	0.9934
	CAO-5	4	0.9929	0.1919	2.1789	0.9929	0.9877	0.3125	2.3714	0.9916
RSO-2 (0–50%)	CAO-1	4	0.9963	0.1011	1.5793	0.9963	0.9940	0.2421	1.8644	0.9949
	CAO-2	4	0.9961	0.1043	1.6064	0.9961	0.9919	0.3026	1.8570	0.9952
	CAO-3	4	0.9956	0.1189	1.7146	0.9956	0.9968	0.1281	2.0092	0.9945
	CAO-4	4	0.9957	0.1164	1.6942	0.9957	0.9952	0.0666	2.0143	0.9939
	CAO-5	4	0.9934	0.1793	2.1059	0.9934	0.9947	0.1306	2.2722	0.9927
RSO-3 (0–50%)	CAO-1	4	0.9896	0.2815	2.6387	0.9896	0.9595	1.3360	3.6806	0.9808
	CAO-2	4	0.9960	0.1088	1.6406	0.9960	0.9818	0.4280	2.1897	0.9932
	CAO-3	4	0.9959	0.1122	1.6662	0.9959	0.9868	0.3926	2.3196	0.9927
	CAO-4	4	0.9942	0.1578	1.9718	0.9942	0.9841	0.3397	2.4511	0.9917
	CAO-5	4	0.9923	0.2082	2.2684	0.9923	0.9811	0.3962	2.7269	0.9897

CAO: camellia oil; RSO: rapeseed oil. RMSEC: root mean square error of calibration, RMSECV: root mean square error of cross validation.

**Table 5 molecules-25-02036-t005:** Major oil sample preparation procedure for experiments.

Calibration Set	CAO	RSO
Calibration sets of CAO adulterated with RSO samples		
1	5 samples (CAO-1 to 5)	5 samples of CAO adulterated with 1% of 3 different RSO (RSO-1 to 3)
2	5 samples (CAO-1 to 5)	5 samples of CAO adulterated with 3% of 3 different RSO (RSO-1 to 3)
3	5 samples (CAO-1 to 5)	5 samples of CAO adulterated with 5% of 3 different RSO (RSO-1 to 3)
4	5 samples (CAO-1 to 5)	5 samples of CAO adulterated with 10% of 3 different RSO (RSO-1 to 3)
5	5 samples (CAO-1 to 5)	5 samples of CAO adulterated with 15% of 3 different RSO (RSO-1 to 3)
6	5 samples (CAO-1 to 5)	5 samples of CAO adulterated with 20% of 3 different RSO (RSO-1 to 3)
7	5 samples (CAO-1 to 5)	5 samples of CAO adulterated with 25% of 3 different RSO (RSO-1 to 3)
8	5 samples (CAO-1 to 5)	5 samples of CAO adulterated with 30% of 3 different RSO (RSO-1 to 3)
9	5 samples (CAO-1 to 5)	5 samples of CAO adulterated with 35% of 3 different RSO (RSO-1 to 3)
10	5 samples (CAO-1 to 5)	5 samples of CAO adulterated with 40% of 3 different RSO (RSO-1 to 3)
11	5 samples (CAO-1 to 5)	5 samples of CAO adulterated with 45% of 3 different RSO (RSO-1 to 3)
12	5 samples (CAO-1 to 5)	5 samples of CAO adulterated with 50% of 3 different RSO (RSO-1 to 3)
Classification sets of CAO and other oil samples		
13	49 samples	7 different CAO, (CAO-1 to 7); 7 different RSO, (RSO-1 to 7); 7 different SBO, (SBO-1 to 7); 7 different CO, (CO-1 to 7); 7 different SFO, (SFO-1 to 7); 7 different PO, (PO-1 to 7); 7 different SO, (SO-1 to7)
Total	229 oil samples (180 oil blends + 49 pure oils)	

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
