# Peer review of "Rapid Classification and Quantification of Camellia (Camellia oleifera Abel.) Oil Blended with Rapeseed Oil Using FTIR-ATR Spectroscopy"

_molecules, 2020, doi:10.3390/molecules25092036_

Round 1
Reviewer 1 Report
The evaluated study describes an interesting and apparently useful method for the qualification and quantification of CAO adulteration, by using ATR-FTIR spectroscopy. Overall the manuscript is well written, the authors provided a large amount of experimental data, the multivariate data analysis is well presented and explained and the implementation of the PLSR calibration strategies is acceptable. Yet, some improvements would be recommended in order to enhance the manuscript and mace it suitable for publication.
1. An impersonal writing style of the abstract would be recommended.
2. Place a measurement unit for the mentioned root mean square errors (abstract, section 2.4, conclusive section).
3. Please consider to define all abbreviations, in the abstract as well as in the body of the manuscript.
4. The manuscript handles a lot of multivariate data analysis, therefore the authors should describe/define the performed techniques (PCA, LDA. PLSR). Add a few paragraphs in the introduction.
5. Enhance the presentation and graphics of Figure 4.
6. Section 3.3. Collection of spectra – Mention the integration time used for the recording of spectra.
7. Section 3.4. lines 348-353 – Why did the authors chose to perform only SG smoothing and derivatives, and not other pre-processing techniques in order to enhance their statistic results and models’ performance?
Reviewer 2 Report
In the Manuscript “Rapid classification and quantification of Camellia (Camellia oleifera Abel.) oil blended with Rapeseed oil using FTIR-ATR spectroscopy”, by Jianxun Han et al., the Authors proposed the use of FTIR spectroscopy coupled to multivariate analysis to evaluate the authenticity of Camellia oil and quantify the adulterant levels of rapeseed oil.
The issue is of current importance, and the proposal of an effective alternative method to tackle the problem of food adulteration is very interesting and useful.
The scientific design of the experimental as well as the support of different multivariate approaches to analyze the data undoubtedly add value to the results.
However, I have a major concern about the fact that the Authors focused - as spectroscopic marker of adulteration - only on the ratio of the intensity at 1119 cm-1 and at 1096 cm-1, respectively assigned to monounsaturated and polyunsaturated acyl groups.
Major point
The Authors based the detection of possible Camellia oil adulteration mainly on the ratio between two peaks, 1119 cm-1 and 1096 cm-1, and they provided an explanation:
“In pure CAO, the band height of 1119 cm−1 was higher than that of 1096 cm−1, which is the opposite of other pure oils (RSO, BSO, CO, RSO, PO and SO). This has also been found between extra virgin olive oil (EVOO) and canola oil [35] as well as olive oil with SFO, CO, and RSO, and cottonseed oil [36]. This shows that these data are in connection with the concentration of monounsaturated and polyunsaturated acyl groups in the oil samples [30]. Oils with a high concentration of monounsaturated acyl groups show large absorbance values for the bands of 1119 cm−1 and low absorbance values for the bands of 1096 cm−1. The opposite is true for the concentration of polyunsaturated acyl groups in the samples, which agrees with the fatty acid composition of vegetable oils shown in Table 2.”
However, I did not understand the assignment of the two marker peaks. From the literatures that they cited, this point is still obscure to me. The Authors have to provide clearer evidence and /or a clear explanation of the proposed assignment, in consideration of the fact that the whole work is based only on the difference of these two bands.
Minor points
- Line 76: “CAO and another six vegetable oils “should be “ …….and other six……)
- Line 141: “It further verified…..” should be corrected
- The assignment of the 1464 cm-1 peak is not unequivocal. It is assigned to hydrocarbon chain CH2 bending and/or CH3 deformation. See for instance:
1) Natalello A, Sasso F, Secundo F: Enzymatic transesterification monitored
by an easy-to-use Fourier transform infrared spectroscopy method. Biotechnol J 2013, 8:133–138.
2) Casal HL, Mantsch HH: Polymorphic phase behaviour of phospholipid membranes studied by infrared spectroscopy. Biochim Biophys Acta 1984, 779:381–401.
Round 2
Reviewer 2 Report
In the revised version of the manuscript, the authors have addressed all my concerns.